# Cytoprotective Compounds in the Primate Eye: Baseline Metabolomic Profiles of *Macaca fascicularis* Ocular Tissues

**DOI:** 10.3390/ijms262210816

**Published:** 2025-11-07

**Authors:** Maxim V. Fomenko, Lyudmila V. Yanshole, Vadim V. Yanshole, Elena Y. Radomskaya, Dmitry V. Bulgin, Renad Z. Sagdeev, Yuri P. Tsentalovich

**Affiliations:** 1Laboratory of Proteomics and Metabolomics, International Tomography Center SB RAS, Institutskaya 3a, Novosibirsk 630090, Russia; m.fomenko@tomo.nsc.ru (M.V.F.); lucy@tomo.nsc.ru (L.V.Y.); vadim.yanshole@tomo.nsc.ru (V.V.Y.); itc@tomo.nsc.ru (R.Z.S.); 2Research Institute of Medical Primatology, Mira Str. 177, s. Vesyoloe, Sochi 354376, Russia; zoomlenazoom@gmail.com (E.Y.R.); molmed1999@yahoo.com (D.V.B.)

**Keywords:** nonhuman primates, metabolomics, NMR, serum, lens, aqueous humor, vitreous humor

## Abstract

Nonhuman primates are often considered as the best animal models for studying human ophthalmological diseases, but the metabolomic composition of primate ocular tissues remains largely unknown. In this work, we performed NMR-based quantitative metabolomic analysis of crab-eating macaque (*Macaca fascicularis*) serum, aqueous (AH) and vitreous (VH) humors, and lens. We determined the concentrations of a total 94 compounds in these tissues, 13 of which play important cytoprotective roles. The obtained metabolomic profiles represent the baseline metabolomes of blood and eye tissues characteristic of young healthy *M. fascicularis* adults. The obtained data indicate that antioxidants ascorbate and ergothioneine are actively pumped from blood into AH with the use of specific transporters, and there is an active transport against the concentration gradient of amino acids from AH into the lens. The comparison of metabolomic profiles of *M. fascicularis* and human ocular tissues shows a very high degree of similarity at the qualitative level, while the quantitative compositions of cytoprotective compounds (antioxidants, osmolytes, and ultraviolet filters) in *M. fascicularis* and human lenses differ. Despite these differences, from the metabolomic viewpoint, *M. fascicularis* are much better models of human diseases than rodents, which are often used in studies of eye disorders.

## 1. Introduction

Currently, study of the pathogenesis of various human diseases is often based on a comparison of transcriptomic, proteomic, and metabolomic profiles of pathological and normal tissues [1,2,3,4,5]. In particular, differences in metabolomic compositions indicate alterations in certain metabolic pathways, which can help to develop disease diagnosis methods and disease treatment strategies. The problem is that some human tissues, in particular ocular tissues, are difficult to obtain. The sampling of control human ocular tissues is often either difficult or impossible, and the majority of human metabolomic ophthalmological studies are based on a use of tissues (lens, cornea, and aqueous (AH) and vitreous (VH) humors) obtained from cadavers several hours post-mortem [6,7,8,9,10], though some of these tissues can also be obtained as post-operational material [6,10,11,12]. Post-mortem tissues might often be unreliable for metabolomic studies, since anaerobic reactions and cell lysis proceeding due to the disruption of the cell energy status and osmoregulation cause fast and strong post-mortem changes in the metabolomic composition of biological tissues and fluids [13,14,15,16,17].

An alternative way of studying human eye diseases is the use of experimental animals. Rodents—mice and rats—are the laboratory animals most often used in ophthalmological studies, for lenticular studies in particular [18,19,20,21]. However, the comparison of metabolomic compositions of human and animal lenses indicate that for human ophthalmological diseases, rodents present rather poor models [8,9,20]. The lens nucleus and its internal cortex consist of fiber cells with minimal metabolic activity, with a high abundance of structural lenticular proteins (crystallins), and without the protein turnover. The crystallins produced in the lens in the prenatal period and early childhood remain in the lens until the end of life. The protection of these long-lived proteins and the fiber cells themselves from oxidative and osmotic stresses and from ultraviolet (UV) radiation almost completely rely on metabolites—antioxidants, osmolytes, and molecular UV filters. These compounds either enter the lens from the surrounding aqueous humor, or are synthesized in the metabolically active lens epithelial cells. Previous studies [8,9,20] show that the sets of cytoprotective compounds in human and rat lenses differ significantly. This indicates that it is extremely desirable to find an animal model with a lens metabolomic composition similar to that of the human lens.

Monkeys and apes are the species genetically closest to humans. Since the anatomical structures and functional capabilities of human and monkey eyes are similar, nonhuman primates are often considered as the best animal models for studying human eye diseases [22]. Primates have been successfully used for studying ocular diseases and development methods of their treatment [23,24,25,26], including amblyopia, cataract, and glaucoma. However, at the present, the metabolomic composition of nonhuman primate ocular tissues remains largely unknown. We found only occasional references on the presence and abundance of individual metabolites in the monkey ocular tissues, such as UV filters in the monkey lens [27,28]. In this work, we obtained the quantitative metabolomic profiles of four tissues from a crab-eating macaque (*Macaca fascicularis*), including the blood plasma, lens, and aqueous and vitreous humors. The obtained quantitative data can be considered as baseline metabolomic profiles of blood and ocular tissues characteristic of young healthy *M. fascicularis* adults, and can be further used for metabolomic studies of the influence of various factors (aging and diseases) on metabolic processes in the body of monkeys. We also compared our results with previously published metabolomic data on human tissues to establish similarities and differences between metabolomes of ocular tissues of humans and nonhuman primates, and to validate the suitability of nonhuman primates as models for human ophthalmological diseases from the biochemical viewpoint.

## 2. Results

### 2.1. Identification and Quantitation of Metabolites in ^1^H NMR Spectra

Examples of the obtained NMR spectra of metabolomic extracts from serum, AH, VH, and lens of *M. fascicularis* with signal assignment are given in Appendix A. In these figures, unrecognized signals are denoted as s109, s151, m232, and so on, where the letter corresponds to the signal multiplicity (“s” for singlet, “d” for doublet, “t” for triplet, and “m” for multiplet), and three digits shows the chemical shift. For example, s151 corresponds to a singlet at 1.51 ppm. In general, the obtained spectra are qualitatively similar to the spectra of human serum, AH, VH, and lens reported earlier by our and other groups [8,11,29,30,31]. Metabolite identification was performed according to their NMR spectra available in databases and in the literature [5,8,9,11,30,31]. We successfully assigned almost all signals in the NMR spectra of serum, AH, and lens. The literature data on VH metabolomic composition and on NMR spectra of compounds present in VH are significantly more limited. For this reason, several signals in the VH spectra (in particular, signals in 4.0–4.6 ppm and 1.9–2.1 ppm regions) remained unassigned. Very likely, these signals correspond to carbohydrates, probably formed due to degradation of hyaluronic acid present in VH in high amounts.

The concentrations of metabolites in the serum, AH, VH, and lens (in µM for fluids and in nmol per gram of wet tissue for the lens) were calculated by the integration of the NMR signals relative to the internal standard DSS followed by normalization to the tissue volume (or weight). In total, together with UV filters (see below), we determined the concentrations of 94 compounds, including 62 compounds in serum, 59 in AH, 45 in VH, and 67 in lens (Figure 1). The obtained quantitative metabolomic data for cytoprotective compounds—osmolytes, antioxidants, and UV filters—are given in Table 1, and the complete set of quantitative metabolomic data is presented in Appendix A. A comparison of the most abundant metabolites in *M. fascicularis* serum, AH, VH, and lens is shown in Figure 2. Previous studies [32] showed that a single extraction according to protocol used in this work recovered approximately 90% of the metabolites. Therefore, the data in Table 1 and Appendix A should be considered as an underestimation of actual metabolite concentrations by 10%.

### 2.2. Identification and Quantitation of Molecular UV Filters by LC-MS and LC-OD

NMR spectra reveal only two UV filters in the *M. fascicularis* lens, 3-hydroxykynurenine *O*-β-d-glucoside (3-OHKG), and kynurenine (KN). To identify and quantify less abundant UV filters, we performed combined LC-OD and LC-MS study of macaque lenses. The UV-Vis chromatogram recorded at 360 nm for the *M. fascicularis* lens extract (Appendix A) is similar to the chromatogram reported for the human lens [33]. The signal assignment was performed according to the mass spectra obtained by LC-MS method with the use of the same column and under the same chromatographic conditions. We identified eight UV filters in the macaque lens: 3-OHKG, KN, 3-hydroxykynurenine (3-OHKN), 3-hydroxykynurenine *O*-β-d-diglucoside (3-OHKDG), 4-(2-amino-3-hydroxyphenyl)-4-oxobutanoic acid *O*-β-d-glucoside (AHBG), 4-(2-amino-3-hydroxyphenyl)-4-oxobutanoic acid *O*-β-d-diglucoside (AHBDG), deaminated 3-OHKG (4-(2-amino-3-hydroxyphenyl)-4-oxocrotonic acid *O*-β-d-glucoside, 3-OHCKAG), and previously not reported UV filter—methylated 3-OHKG (Me-3-OHKG, C_17_H_24_N_2_O_9_). Three detected UV filters remained unidentified. The major properties of UV filter compounds have been described in previous publications [33,34,35,36,37]. The quantification of UV filters was performed by their absorption spectra, assuming that all kynurenine-derived molecules have the same absorption coefficients at the UV absorption maximum near 360 nm, ε (360) = 4.5 × 10^3^ M^−1^ cm^−1^ [9,38]. The only exception is 3-OHCKAG, for which the value ε (410) = 1.7 × 10^3^ M^−1^ cm^−1^ was taken [9,34]. The most abundant UV filter in the *M. fascicularis* lens is 3-OHKG, which accounts to approximately 90% of the total UV filter absorbance at 360 nm, followed by AHBG and KN.

### 2.3. Comparison of In Vivo and Post-Mortem Blood Samples

Samples of blood, lens, AH, and VH from *M. fascicularis* were taken 15–20 min post-mortem. The goal of the preliminary stage of the study was to estimate the scale of post-mortem metabolomic changes occurring in these tissues. For this purpose, we compared the metabolomic profiles of *M. fascicularis* blood taken in vivo (seven individuals) and post-mortem (six individuals). It is known [13,15,16,39,40] that the primary post-mortem event affecting blood metabolomic composition is the disruption of the majority of energy-dependent metabolic processes, TCA cycle, purine catabolism, and functioning of Na^+^/K^+^ pumps, in particular. The disruption of TCA cycle and purine catabolism causes fast decay of pyruvate and significant elevation of fumarate, succinate, and hypoxanthine levels [13,15]. The dysfunction of Na^+^/K^+^ pumps leads to the increase in the intracellular osmotic pressure, cell lysis, and cytoplasm leakage into intercellular space, greatly elevating the levels of amino acids and other metabolites in the blood plasma [15,16,40,41]. In our measurements, we found that in post-mortem samples, the concentration of pyruvate decreased almost threefold, while the levels of fumarate, succinate, hypoxanthine, and xanthine increased by factors of 2.5, 2, 1.7, and 2.7, respectively (Appendix A). At the same time, concentrations of the majority of amino acids did not change. These observations indicate that under our conditions, the post-mortem metabolomic changes in the blood are at the very beginning, and the cell lysis has not started yet. In recent reports [15,16], we demonstrated that the post-mortem changes in the ocular tissues proceed significantly more slowly than in the blood. Therefore, one can assume that the quantitative metabolomic compositions of AH, VH, and lens of *M. fascicularis* reported in the present work are very close to the in vivo metabolomic profiles of these tissues.

### 2.4. Serum Metabolomic Composition

The major constituents of the *M. fascicularis* serum are amino acids and organic acids; they represent 43 out of 62 compounds detected in the serum in the present work. The most abundant amino acids are proteinogenic amino acids alanine, glutamine, glycine, and lysine with concentrations between 200 and 700 µM. Among all metabolites, the highest level was found for lactate (up to 20 mM) followed by glucose (4–6 mM). We also detected small quantities of nitrogenous bases and their derivatives, including allantoin, AMP, creatinine, hypoxanthine, inosine, uridine, and xanthine. Typically, these compounds are abundant in cytoplasm, and their presence in the blood plasma might be at least partly attributed to the beginning of post-mortem processes occurring during either the post-mortem interval (for post-mortem samples) or the sample preparation (for in vivo samples).

### 2.5. AH Metabolomic Composition

Qualitatively, the metabolomic profiles of the blood serum and AH measured in the present work are rather similar; however, we observed some distinct quantitative differences. In particular, AH contains much lower levels of glycine, acetate, glucose, mannose, and betaine, while the concentrations of methionine, pyroglutamate, GABA, malonate, pyruvate, *myo*-inositol, ascorbate, ergothioneine, 3-OHKG, and AMP are much higher.

### 2.6. Lens Metabolomic Composition

For majority of proteinogenic amino acids, their levels in the lens are approximately twofold higher than that in AH. One should take into account that the structural lens proteins, crystallins, constitute about 40% of the lens tissue. Therefore, one can conclude that the concentrations of amino acids in the protein-free intracellular space of the lens is at least three- or fourfold higher than in AH. High amino acid demand in the lens should be attributed to the non-stop synthesis of crystallins in differentiating lens cells. Most likely, the elevated amino acid concentrations in the lens are produced by transport proteins pumping amino acids from AH against the concentration gradients [42,43,44].

An important feature of the lens metabolome is a high abundance of compounds providing the cellular protection: osmolytes, antioxidants, and UV filters. In the *M. fascicularis* lens, the major osmolyte is *myo*-inositol, the major UV filter—3-OHKG, and the major antioxidants—GSH, ascorbate, and ergothioneine.

### 2.7. VH Metabolomic Composition

The composition of amino acids in the *M. fascicularis* VH is poorer than that in other ocular tissues; we reliably detected and quantified only 11 proteinogenic amino acids, and their concentrations are lower than that in serum, AH, and lens. The most abundant metabolites in VH are lactate and glucose, which constitute more than 70% of the total metabolite abundance, followed by glutamine and threonate. As we mentioned above, a number of signals in the NMR spectrum of VH correspond to unknown compounds. Previous NMR studies of rabbit, porcine, and human VH [15,16,45] do not help to assign these signals, and their correct attribution requires a separate study. This study should include chromatographic separation of metabolomic extracts from VH into fractions, each of which contains only a few metabolites, followed by NMR and MS characterization of unknown compounds. We plan to conduct such a study in the near future.

## 3. Discussion

In this work, we have measured the quantitative metabolomic compositions of *M. fascicularis* serum and ocular tissues, AH, VH, and lens. Earlier, the metabolomic studies were performed for biological fluids of nonhuman primates, including blood, urine, milk, sperm, and cerebrospinal fluid [46,47,48,49,50,51,52,53], and the data for blood are in a good agreement with our results. Detailed quantitative metabolomic profiles of ocular tissues are published in this work for the first time.

### 3.1. Comparison of M. fascicularis Serum and AH Metabolomic Profiles

The Volcano plot in Figure 3 shows the comparison of metabolomic compositions of *M. fascicularis* AH and serum. AH is secreted by the ciliary processes through both the active secretion and the passive diffusion/ultrafiltration of blood plasma [54], and the metabolomic compositions of plasma and AH are rather similar [11].

The difference between the metabolite concentrations in serum and AH is determined by three major factors: (a) An active pumping into AH of metabolites playing key roles in the homeostasis of ocular tissues, the eye lens in particular; these pumps are located at the blood-ocular barrier; (b) a fast metabolite consumption by intraocular tissues, leading to the decrease in their concentrations in AH as compared to blood; (c) a metabolite secretion from ocular tissues into AH leading to the increase in their concentrations in AH as compared to blood. The comparison of the metabolomic data for serum and AH shows that the levels of several compounds in AH is much higher, including ascorbate, ergothioneine, 3-OHKG, *myo*-inositol, pyroglutamate, GABA, and AMP. Ascorbate and ergothioneine are exogenous antioxidants; the cells of primates do not synthesize these compounds, and primates obtain them only with food. However, the level of ergothioneine in AH (46 nmol/g) is higher than that in serum (3 nmol/g) by an order of magnitude; the concentration of ascorbate in AH is 590 nmol/g, while in the blood it is below the NMR detection limit. These observations clearly indicate an active transport of ascorbate and ergothioneine into AH. The pumping of ascorbate from blood into AH has been shown many years ago [11,55,56]; here, we demonstrate that the same mechanism operates for ergothioneine as well. Earlier, the existence of the ergothioneine-specific transporter (OCTN1) was discovered by Gründemann et al. [57], and the accumulation of ergothioneine in the eye was shown for mice [58,59,60]. The existence of the mechanism of the ergothioneine accumulation inside the eye indicates the importance of this compound for normal eye functioning. In contrast, UV filter 3OHKG and osmolyte *myo*-inositol are produced in the metabolically active lens epithelial monolayer and the outer cortex. The concentrations of these compounds in the lens are very high (Table 1), and their presence in AH should be attributed to the diffusion from the lens. Pyroglutamate, GABA, and AMP participate in many metabolic cycles, and their enhanced levels in AH as compared to blood may originate from the diffusion from different ocular tissues, including the lens.

The levels of glycine, glucose, mannose, and betaine in AH are much lower than in blood (Appendix A). These metabolites are actively used in intraocular tissues for energy production and for intracellular synthesis, and their low concentrations in AH can be attributed to the fast consumption by the lens and other ocular tissues.

### 3.2. Comparison of Metabolomic Profiles of Human and M. fascicularis Tissues

#### 3.2.1. Metabolomic Profiles of Serum

The similarity of metabolomic compositions of human and nonhuman primate serum has been shown in previous publications [50], and the data obtained in the present work confirm this statement. The comparison of metabolomic data on *M. fascicularis* serum (Appendix A) with the published data on human serum [5] shows that the concentrations of the majority of amino acids, organic acids, sugars, amines, and other compounds are similar. We found significant differences for only a few metabolites. In particular, the levels of lactate, *myo*-inositol, and *N*-acetylcarnitine in *M. fascicularis* serum are significantly higher. This is also in a good agreement with the previous report [50].

#### 3.2.2. Metabolomic Profiles of AH

With the metabolomic difference between human and *M. fascicularis*, AH is somewhat more pronounced. We compared the data obtained in the present work (Table 1 and Appendix A) with the recently published [10] AH metabolomic profile of young (average age of 31 years) human adults. In that work, AH samples were collected in vivo during corneal transplantation operations. Significant (fold change > 2, *p* < 0.01 (FDR)) difference was found for 11 metabolites (Figure 4). Among them, two metabolites have higher abundance in the human AH, and nine in the *M. fascicularis* AH. It should be noted that the levels of *myo*-inositol and 3-OHKG in the *M. fascicularis* lens is much higher than that in the human lens, and the higher abundance of these compounds in *M. fascicularis* AH as compared to human AH most likely corresponds to their diffusion from the lens.

#### 3.2.3. Metabolomic Profiles of Lens

From a biomedical viewpoint, the comparison of *M. fascicularis* and human lenses is the most interesting one. Due to the ethical restrictions, normal human lenses with the short post-mortem interval (PMI) are practically unavailable, and the majority of ophthalmological experimental studies (especially the cataract-related studies) are performed with the use of experimental animals, usually rodents. For that reason, we decided to compare the metabolomic profile of *M. fascicularis* lens with the previously obtained metabolomic data on the human and rat eye lenses [9,20]. The data in the paper [9] are given for normal human lenses with the post-mortem intervals of 7–17 h. Apparently, significant metabolomic changes did occur during the PMI, including the depletion of energy metabolites, lactate accumulation, changes in concentrations of compound from the TCA cycle, and elevation of amino acid levels due to the protein hydrolysis [13,15,16,39,40,61]. Therefore, we restricted our comparison to compounds protecting the lens tissue against the cataract development.

Cataract is the most common age-related eye disease. It is generally accepted that the main cause of the cataract development is the oxidative stress leading to the post-translational modifications of crystallins, their insolubilization, aggregation, and precipitation [62]. Violation of the correct laying of fiber cells in the lens may also contribute to the cataractogenesis [63]. Due to very low metabolic activity in the lens nucleus, the defense of the lens cells against UV radiation, oxidative, and osmotic stresses rely on cytoprotective molecules—UV filters, antioxidants, and osmolytes. Figure 5 shows the graphical representation of the cytoprotective compound compositions in the human, *M. fascicularis*, and rat lenses. Qualitatively, the sets of cytoprotective metabolites in the human and *M. fascicularis* lenses are very similar, and differ significantly from that in the rat lens. The main antioxidants in the both human [9] and *M. fascicularis* lenses are reduced glutathione (GSH), ascorbate, and ergothioneine, while in the rat lens, only GSH is present at the significant level [20], and its abundance is approximately threefold higher than that in the human lens. The major osmolytes in the rat lens are taurine and hypotaurine, while in the human and *M. fascicularis* lenses it is *myo*-inositol. The human and *M. fascicularis* lenses contain high concentrations of 3-OHKG and other kynurenine derivatives acting as UV filters, while the rat lens does not contain UV filters.

Nevertheless, quantitatively, the metabolomes of the human and *M. fascicularis* lenses differ. The levels of ascorbate and GSH in the human eye lens are almost an order of magnitude higher than that in the *M. fascicularis* lens, while the concentration of ergothioneine is almost an order of magnitude lower. The abundance of the major lens osmolyte, *myo*-inositol, in the *M. fascicularis* lens is almost three times higher than that in the human lens, and the level of the major UV filter, 3-OHKG, is approximately five times higher. This difference, at least in part, could be attributed to the age-related factor. It has been reported that 3-OHKG concentration in the human lens decreases with age [33], and *myo*-inositol concentration decreases in rat [64] and presumably in human lens [65,66]. The human lenses in work [9] were from individuals after middle age, while the lenses studied in the present work were taken from young (four-year-old) monkeys. In addition to that, the human lens contains a substantial amount of GSH-3-OHKG—a product of the GSH addition to 3-OHCKAG [33,35,67]—which, in turn, is the product of 3-OHKG deamination [34,68,69]. In the *M. fascicularis* lens, we did not detect this compound. This difference can be attributed to two major factors. First, the formation of GSH-3-OHKG in the human lens occurs mostly after middle age [33], again pointing at the age-related factor. Perhaps, in lenses of old macaques, GSH-3-OHKG is also present in detectable amounts. Second, the GSH level in the *M. fascicularis* lens is much lower than that in the human lens, so the probability to form GSH-3-OHKG adduct is lower. Nevertheless, it is difficult to judge from the data obtained whether the observed quantitative metabolomic differences between *M. fascicularis* and human lenses should be attributed to the different ages, lifestyles, and feeding behaviors, or if they are genetically fixed.

## 4. Materials and Methods

### 4.1. Sample Collection

All experiments involving animals were performed in accordance with the ARVO Statement for the Use of Animals in Ophthalmic and Vision Research and the European Union Directive 2010/63/EU on the protection of animals used for scientific purposes, and approved by bioethics committee of the Research Institute of Medical Primatology (Record 80/1 from 2 December 2021).

AH, VH, lens, and plasma were obtained from six 4-year-old crab-eating macaque (*Macaca fascicularis*) species (3 female and 3 male); in vivo plasma was obtained from seven other 4-year-old species (3 female and 4 male). In vivo and post-mortem blood samples were collected by venipuncture using a syringe with a 21G needle. Within 10–15 min after the blood collection, blood samples were centrifuged (3000× *g*, 10 min), and the plasma obtained was transferred into separate vials. The plasma samples were immediately frozen and kept at −70 °C until analysis.

Euthanasia was performed by intravenous administration of 5.0 mL of Anestofol 5% (Interpharm, Moscow, Russia) with preliminary general anesthesia by intravenous injection of 0.10 mL/kg of Xila (2% xylazine) (Interchemie Werken “de Adelaar” BV, Castenray, The Netherlands) and 0.05 mL/kg of Zoletil^®^ (Virbac Sante Animale, Carros, France). The samples of AH were taken from the anterior chamber by the cornea puncture with a 21G needle. Then, the cornea was cut along the iris border, the lens was extracted, and VH was collected with the 21G needle. The samples of AH, VH, and lens were frozen and kept at −70 °C.

### 4.2. Sample Preparation

Sample preparation for all biological fluids and tissues was performed according to the procedures described in refs [15,16,61]. Typical lens weight was 90–100 mg, typical plasma volume was 300 µL, VH volume was 600 µL, and AH volume was 70–200 µL. Lenses were placed in glass vials and homogenized with a TissueRuptor II homogenizer (Qiagen, Venlo, The Netherlands). Biological fluids and lens homogenate were subjected to liquid–liquid extraction in methanol–chloroform–water mixture. Metabolomic extracts were collected, divided into two parts for NMR (2/3) and LC–MS (1/3) analyses, and lyophilized.

### 4.3. NMR Measurements

^1^H NMR spectra were obtained as described in Ref. [61]. Briefly, we re-dissolved the extracts for NMR measurements in 600 μL of D_2_O containing 2 × 10^−5^ M sodium 4,4-dimethyl-4-silapentane-1-sulfonic acid (DSS) as an internal standard and 50 mM deuterated phosphate buffer (pH 7.2). We performed the NMR measurements at the Center of Collective Use “Mass spectrometric investigations” SB RAS with the use of an NMR spectrometer AVANCE III HD 700 MHz (Bruker BioSpin, Ettlingen, Germany) equipped with a 16.44 Tesla Ascend cryomagnet. The spectra were obtained with the use of zgpr sequence with the water signal suppression, 10 s relaxation delay, 6.7 s acquisition time, 70° detection pulse, and 64 scans. Preliminary measurements showed that for most metabolites, the T1 relaxation time ranged from 1.5 to 4 s. The longest relaxation times were found for histidine (7.6 s) and acetate (6.2 s) protons. Therefore, under our experimental conditions (total scan time 16.7 s, and 70° detection pulse) correct concentration values can be expected for all metabolites.

### 4.4. LC-MS and LC-OD Measurements

We performed liquid chromatography with mass spectrometric (LC-MS) and UV-Vis (LC-OD) measurements simultaneously on two instrumental setups with the same HPLC conditions. The extracts for LC-MS and LC-OD analyses were re-dissolved in 100 μL of 10 mM ammonium formate and 0.1% formic acid (*v*/*v*) solution in H_2_O. The LC separation was performed on an Intensity Solo 2 C18 column (2 × 100 mm, 2 µm, 100 Å, Bruker Daltonics) with a pre-column. The LC conditions were as follows: mobile phase A—10 mM ammonium formate in H_2_O, mobile phase B—acetonitrile, both with the presence of formic acid 0.1%; flow 0.2 mL/min; the gradient was 5% B (0–3 min), 5–95% B (3–20 min), 95% B (20–25 min), 95–5% B (25–25.5 min), and 5% B (25.5–35 min); the injection volume was 10 μL.

The LC-OD setup consisted of an LC 1200 HPLC system (Agilent Technologies, Santa Clara, CA, USA), equipped with a quaternary pump, an autosampler, and a diode array detector (DAD). The chromatograms were monitored at 220, 260, 280, 360, and 450 nm and the analysis of LC-OD data was performed using either ChemStation revision B.04.03 software (Agilent Technologies, Santa Clara, CA, USA) for these fixed wavelengths or OpenChrom v.1.5.0 software (Lablicate GmbH, Hamburg, Germany) for other wavelengths from LC-DAD data.

The LC-MS setup consisted of an Elute UHPLC chromatograph (a binary pump with HPG, an autosampler, and a column thermostat) with an Impact II high-resolution electrospray quadrupole time-of-flight (ESI-Q-TOF) mass spectrometer (Bruker Daltonics, Germany). Mass spectra were recorded in the positive ion mode over the 50–1500 *m*/*z* range with 3 Hz scan rate. Each LC-MS chromatogram contained a calibration segment in the void, where sodium formate clusters used as a calibrant were supplied by a syringe pump connected to an ESI source via a divert valve. The typical resolution was ca. 30,000 and the accuracy <1 ppm. The data obtained were analyzed using DataAnalysis 5.3 software (Bruker Daltonics, Germany).

LC-MS and LC-OD setups were used only for identification and quantification of UV filters. The quantification was performed by their absorption at 360 nm using kynurenine as a chemical standard, assuming that these kynurenine derivatives have the same absorption coefficients at 360 nm (ε_360_ = 4.5 × 10^3^ M^−1^ cm^−1^).

### 4.5. Data Analysis

The phase and baseline corrections as well as signal integration in NMR spectra were performed using MestReNova v.12 software (Mestrelab Research S.L., A Coruna, Spain). Metabolite identification was based on chemical shifts and multiplicities for individual compounds obtained from the HMDB (https://hmdb.ca, accessed on 13 November 2023) and AMDB (https://amdb.online, accessed on 13 November 2023) databases, Chenomx NMR Suite v.8 (Chenomx Inc., Edmonton, AB, Canada), and from our previous reports on lens and other tissues’ metabolomic composition [8,9,16,20,61]. The concentrations of metabolites in the samples were determined by the peak area integration relative to the internal standard DSS, and then recalculated into metabolite concentrations in the tissue (for the lens, in nmoles per gram of the tissue wet weight, and for AH, VH, and serum, in nmoles per milliliter).

Statistical treatment of metabolomics data was performed at the MetaboAnalyst 5.0 web-platform (https://www.metaboanalyst.ca, accessed on 13 November 2023, [70]). A Volcano plot was built with the following parameters: Mann–Whitney *U*-test with the false discovery rate (FDR) correction of *p*-values, and the threshold for *p*-value < 0.01 and fold change (FC) above 2.

## 5. Conclusions

The results obtained in the present work indicate that the metabolic processes in the ocular tissues of humans and nonhuman primates proceed in a similar way. First, the qualitative metabolomic compositions of such tissues as serum, AH, and lens are very similar. In particular, the set of lenticular cytoprotective compounds—antioxidants, osmolytes, and UV filters—are practically the same for both humans and macaques. Second, there is an active transport of exogenous antioxidants ascorbate and ergothioneine from blood into AH, greatly enhancing their concentrations in the ocular tissues. Third, the concentrations of the majority of proteinogenic amino acids in the lenses of humans and *M. fascicularis* are significantly higher than that in AH. This indicates an active pumping of amino acids into lens against the concentration gradient for both species. Finally, similar combinations of kynurenine-like molecules show that the mechanisms of the synthesis and degradation of UV filters in the lens of human and nonhuman primates are the same. Therefore, from a biochemical viewpoint, nonhuman primates are excellent models for studying human ophthalmological diseases.

## Figures and Tables

**Figure 1 ijms-26-10816-f001:**
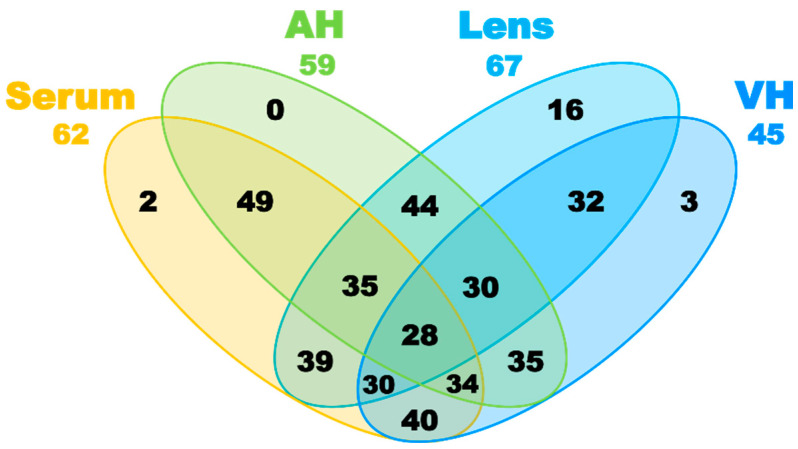
Venn diagram for metabolites detected in the *M. fascicularis* serum, AH, VH, and lens.

**Figure 2 ijms-26-10816-f002:**
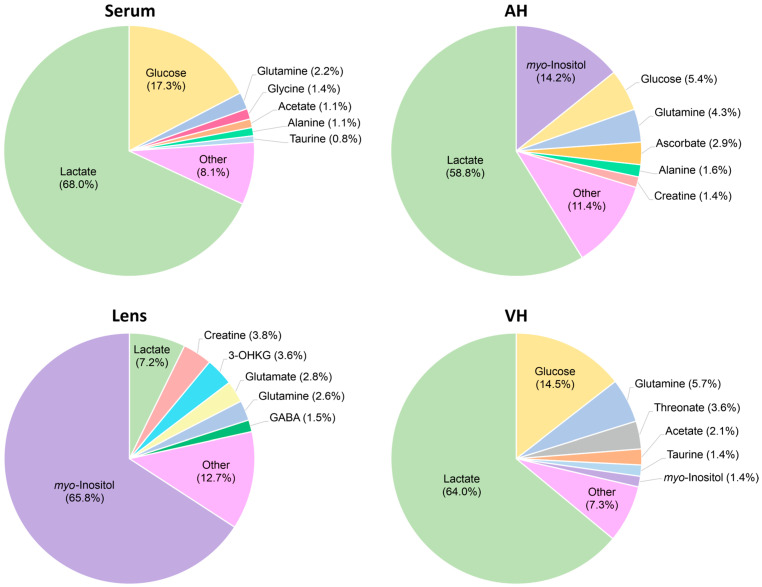
Most abundant metabolites in the *M. fascicularis* serum, AH, lens, and VH.

**Figure 3 ijms-26-10816-f003:**
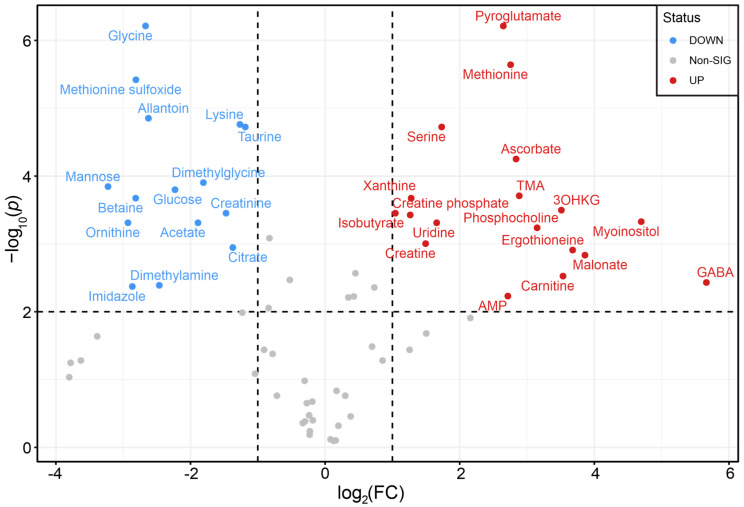
Volcano plot for metabolites in *M. fascicularis* AH and serum. Thresholds are indicated as dashed lines for the FC > 2 and *p* < 0.01 (FDR-corrected). Red dots indicate higher metabolite concentration in AH, blue dots indicate higher concentrations in serum.

**Figure 4 ijms-26-10816-f004:**
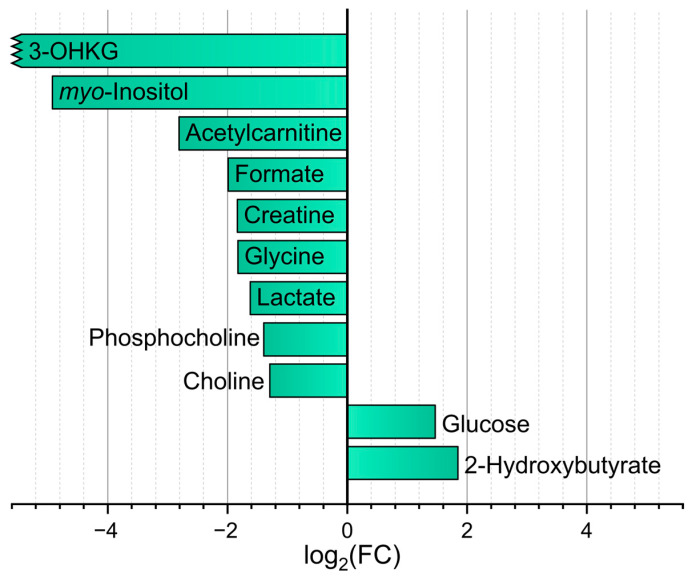
Approximate ratio of metabolite concentrations for compounds with statistically significant difference (FC > 2, *p* < 0.01 (FDR)) in human [11] to that in macaque (this work) AH. Bars to the left indicate higher levels in macaque, and to the right, in human. Jagged bar ends indicate ratio > 30.

**Figure 5 ijms-26-10816-f005:**
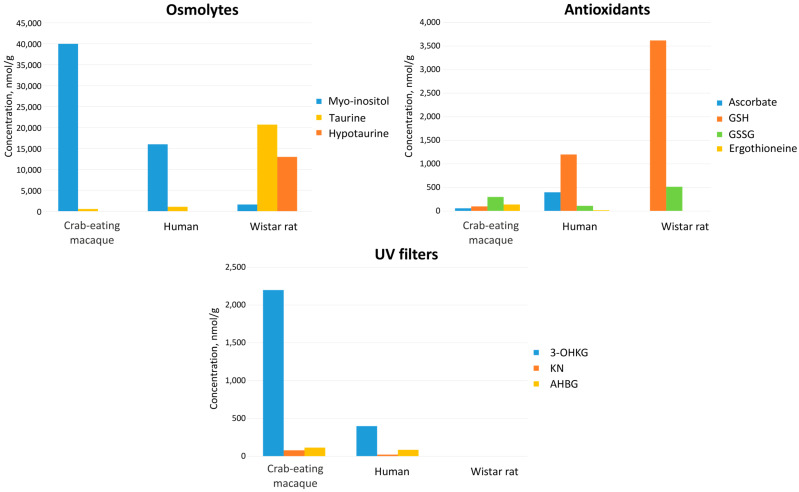
Comparison of levels of osmolytes, antioxidants, and UV filters in the lenses of macaque (this work), human [9], and rat [20].

**Table 1 ijms-26-10816-t001:** Metabolite concentrations in *M. fascicularis* serum, AH, lens, and VH. Values are presented as mean ± standard deviation (SD).

Metabolite	Serum In Vivo, µM	Serum Post-Mortem, µM	AH Post-Mortem, µM	Lens Post-Mortem, nmol/g	VH Post-Mortem, µM
Osmolytes					
Taurine	109 ± 21	246 ± 25	108 ± 24	570 ± 160	190 ± 190
*myo*-Inositol	92 ± 11	110 ± 40	2900 ± 1100	40,000 ± 5000	180 ± 40
Antioxidants					
Ascorbate	ND ^1^	ND	590 ± 150	63 ± 25	83 ± 9
Ergothioneine	0.7 ± 1.0	3.0 ± 2.9	46 ± 21	140 ± 60	ND
GSH	ND	ND	ND	83 ± 22	ND
GSSG	ND	ND	ND	320 ± 70	ND
Molecular UV filters					
3-OHKG	ND	ND	160 ± 60	2200 ± 700	ND
3-OHCKAG ^2^	ND	ND	ND	1.8 ± 0.9	ND
3-OHKDG ^2^	ND	ND	ND	0.7 ± 0.4	ND
3-OHKN ^2^	ND	ND	ND	4.9 ± 1.1	ND
AHBDG ^2^	ND	ND	ND	5.8± 2.8	ND
AHBG ^2^	ND	ND	ND	109 ± 18	ND
KN	ND	ND	ND	80 ± 50	ND
Me-3-OHKG ^2^	ND	ND	ND	9.4 ± 1.1	ND

^1^ ND—not detected. The value is below the NMR limit of quantification (LOQ). ^2^ Concentration value is determined by LC-OD method.

## Data Availability

Raw NMR spectra, description of specimens and samples, metabolite concentrations, and the preliminary metabolomic analysis is available at our Animal Metabolite Database repository, Experiment ID 273 (https://amdb.online/amdb/experiments/273/, accessed on 7 September 2025) and at the MetaboLights repository, study identifier MTBLS8943 (https://www.ebi.ac.uk/metabolights/MTBLS8943/, accessed on 7 September 2025). All other data are available from the corresponding author upon request.

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
