# Peer review of "Cytoprotective Compounds in the Primate Eye: Baseline Metabolomic Profiles of *Macaca fascicularis* Ocular Tissues"

_ijms, 2025, doi:10.3390/ijms262210816_

Round 1
Reviewer 1 Report
Comments and Suggestions for Authors
The heart of this study relies on NMR spectroscopy. Authors perform the assignment of metabolites based on NMR databases. However every matrix is different and metabolites can have a shift respect those databases, therefore everytime a new matrix is studied some check should be performed. Of course there are peaks that can be assigned unambigosuly by its fingerprint, chemical shift, multiplicities... but there are many which are difficult, due to low concentration, overlapping, poor digital fingerprint (sometimes just defined with a single singlet), and probably they are assigned just temptatively. In fact normally to increase the degree of assignment confidence authors should perform a complete set of 2D NMR experiments at least for a representative sample (COSY, HSQC, JRES...), or even if more reliability is still needed spiking methodology should be performed carefully. The task is huge, so if not posible to perform this spiking task by authors, at least they clearly should specify for each metabolite their degree of confidence at a qualitatively level as high, medium or low.
On the other hand, authors claim they performed quantitative analysis only using area integration. Which NMR experimental conditions did they perform?, did they use quantitative conditions (long recycle delay, 90 pulse...). Please, specify 1H-NMR experimental conditions.
Anyway, quantification is very difficult to achieve in such complex matrices. Basically, I don't believe they can really integrate all the set of presented metabolites in a free region (without overlapping). Probably, some peaks are isolated, but for sure not all of them. So, another confidence list should be provided with the metabolites that clearly are presenting isolated signals (quantitative analyisis) or those presenting signal overlap (semiquantitative analyisis). Also another point is that there are metabolites presenting very low signal intensities, almost at noise level and clearly below the limit of quantification (LOD), those should be also specified as semiquantitative analyisis.
To improve quantitative analysis in overlapping regions authors should perform deconvolution with specialized software like Chenomx, if not posible authors should at least be less confident along the text about quantitative NMR.
Reviewer 2 Report
Comments and Suggestions for Authors
The paper “Cytoprotective Compounds in the Primate Eye: Baseline Metabolomic Profiles of Macaca fascicularis Ocular Tissues”
conducted by Maxim V. Fomenko et al. provide very important and detailed informations regarding the similarity of metabolic processes in human and non-human primate ocular tissues.
The basic problem was to identify a similarity between the metabolic processes that occur in human and non-human primate ocular tissues, considering the possibility of using them in control studies in ophthalmology, given the difficulty of sampling human ocular tissues.
The results showed the similarity of the qualitative metabolomic compositions of some tissues as well as of some lenticular cytoprotective compounds in humans and macaques. As a result of the similar biochemical composition, non-human primates can be considered excellent models for studying human ophthalmological diseases.
An important aspect highlighted by this research is related to the detailed quantitative metabolomic profiles of ocular tissues, which are published in this work for the first time.
The formulated conclusions are in accordance with the obtained results, based on the proposed study objectives.
The references are recent and relevant to the information presented.
Author Response
Dear Reviewer,
Thank you for taking the time to read our article and for your positive review of this work.